# Myriad: a real-world testbed to bridge trajectory optimization and deep learning

**Nikolaus H. R. Howe**
Mila, Université de Montréal
`niki.howe@mila.quebec`

**Simon Dufort-Labbé**
Mila, Université de Montréal

**Nitarshan Rajkumar**
University of Cambridge[*]

**Pierre-Luc Bacon**
Mila, Université de Montréal, Facebook CIFAR AI, IVADO

## Abstract

We present Myriad, a testbed written in JAX which enables machine learning researchers to benchmark imitation learning and reinforcement learning algorithms against trajectory optimization-based methods in real-world environments. Myriad contains 17 optimal control problems presented in continuous time which span medicine, ecology, epidemiology, and engineering. As such, Myriad strives to serve as a stepping stone towards application of modern machine learning techniques for impactful real-world tasks. The repository also provides machine learning practitioners access to trajectory optimization techniques, not only for standalone use, but also for integration within a typical automatic differentiation workflow. Indeed, the combination of classical control theory and deep learning in a fully GPU-compatible package unlocks potential for new algorithms to arise. We present one such novel approach for use in optimal control tasks. Trained in a fully end-to-end fashion, our model leverages an implicit planning module over neural ordinary differential equations, enabling simultaneous learning and planning with unknown environment dynamics. All environments, optimizers and tools are available in the software package at `https://github.com/nikihowe/myriad`.

## 1 Introduction

The rapid progress of machine learning (ML) algorithms is made clear by the yearly improvement we see on standard ML benchmarks (Deng et al., 2009; Todorov et al., 2012; Bellemare et al., 2013). Inevitably, the popularity of a given testbed creates a positive feedback effect, encouraging researchers to develop algorithms that achieve good performance on that set of tasks (Kerner, 2020; Henderson et al., 2018). We believe it is crucial that our algorithms be well-suited for positive-impact, real-world applications. As such, we must be able to train and test them on real-world-relevant tasks.

To this end, we present Myriad, a real-world testbed for optimal control methods such as imitation learning and reinforcement learning (RL). Myriad differs from previous testbeds in several key aspects. First and most importantly, all tasks are inspired by real-world problems, with applications in medicine, ecology, epidemiology, and engineering. Second, Myriad is, to our knowledge, the first repository that enables deep learning methods to be combined seamlessly with traditional trajectory optimization techniques. Figure 1 shows a visualization of applying a one such trajectory optimization technique on an environment implemented in Myriad. Third, the system dynamics in Myriad are continuous in time and space, offering several advantages over discretized environments. On the one hand, these algorithms are adaptable to changing sampling frequencies, or even irregularly spaced

---

[*]Work done while at Mila, Université de Montréal.

36th Conference on Neural Information Processing Systems (NeurIPS 2022) Track on Datasets and Benchmarks.

data. At the same time, using continuous-time dynamics gives the user freedom to choose between integration techniques, and opens the door to efficient variable-step integration methods. These can take advantage of the local environment dynamics to effectively trade off speed and accuracy of integration (Fehlberg, 1969).

While the field of control theory has yielded practical approaches to solve a plethora of problems in the industrial setting (Lenhart and Workman, 2007; Betts, 2010; Biegler, 2010), adoption of such methods within the ML community has been limited. This is in part due to the historical focus of ML on scalability, and of control theory on robustness and optimality, and is further exacerbated by the lack of tools to run control algorithms in a deep learning setting. Yet trajectory optimization techniques can offer excellent performance in systems with known dynamics, and thus serve as a solid benchmark against which to test RL techniques. Additionally, trajectory optimization offers several advantages over standard RL, such as the ability to impose safety constraints, which is crucial in real-world applications (Betts, 2010; Biegler, 2010). As such, the Myriad repository allows for the combination of trajectory optimization and deep learning techniques into powerful hybrid algorithms. As an example, we implement an end-to-end trained implicit planning imitation learning algorithm, and benchmark its performance alongside that of trajectory optimization on known and learned dynamics models.

**Figure 1:** The optimal trajectory of pesticide use over time and resulting population dynamics in the Predator Prey domain, which is included in Myriad (see Table 1 for a complete list of environments). Direct single shooting (see Section 4) was used to compute the optimal control trajectory.

The following sections present various aspects of Myriad, starting with a review of related work in **Section 2**. In turn, **Section 3** gives an overview of the Myriad repository and describes several of the available control environments. The subsequent sections can be thought of as both presenting the tools in Myriad, as well as describing the building blocks used to create the aforementioned imitation learning algorithm. To start, **Section 4** describes direct single shooting, a standard trajectory optimization technique, and **Section 5** shows how we can leverage GPU-accelerated first-order methods to use trajectory optimization in a machine learning setting. **Section 6** presents the system identification problem, and how Myriad can be used to learn neural ordinary differential equation models (Chen et al., 2018) of unknown system dynamics. Finally, **Section 7** presents a new deep imitation learning algorithm which includes a control-oriented inductive bias, trained end-to-end using tools from Myriad. **Section 8** concludes and discusses the limitations and potential societal impact of this work.

We summarize our main contributions as follows:

- We present a **testbed for real-world tasks**, including learning dynamics models from data and the problem of optimal control. The testbed contains 17 continuous-time real-world tasks, and it is straight-forward to add additional systems to the repository.
- We provide a set of plug-and-play **differentiable trajectory optimization algorithms** implemented in JAX (Bradbury et al., 2018), which can be used standalone or in conjunction with machine learning techniques.
- We introduce a **novel control-oriented imitation learning algorithm** which combines optimal control with deep learning. The tools in Myriad enable us to develop this method and compare its performance to control methods which leverage known or learned dynamics.
- We collect **benchmark reference scores** for most environments, achieved using classical optimal control techniques with the true system dynamics, as well as benchmark scores achieved using trajectory optimization on two kinds of learned dynamics model. See Appendix C for these scores along with details on how they were obtained.

## 2 Related work

**Testbed**: Within the context of RL, several testbeds have been influential in pushing forward the state-of-the art, notably OpenAI Gym (Brockman et al., 2016) and the Arcade Learning Environment (Bellemare et al., 2013). Yet, these environments are inherently discrete in time, and focus primarily on game-like settings, abstracting away much of the challenge when working with real-world problems. There also exists a rich collection of software for robotics tasks, some of which are differentiable (Tedrake et al., 2019; Todorov et al., 2012; Coumans and Bai, 2021). However, these are narrow in scope, only focusing on physics simulation, which makes them challenging to leverage to build a testbed for other kinds of real-world problems.

While there exist ML testbeds for real-world tasks (Koh et al., 2021), Myriad is to our knowledge the first to focus on learning and control, and to leverage trajectory optimization in a deep learning context. Indeed, even packages which provide real-world trajectory optimization problems often rely on symbolic differentiation and non-differentiable solvers to compute optimal trajectories (Antony, 2018; Andersson et al., 2019), making them unsuitable for use in a deep learning workflow.

**Algorithm**: Various attempts have been made to include optimal control techniques within a larger neural network architecture, for example by creating a differentiable sub-network (Gould et al., 2016) with gradients computed via loop unrolling (Okada et al., 2017; Pereira et al., 2018) or implicit differentiation (Mairal et al., 2011; Amos and Kolter, 2017; Amos et al., 2018; Jin et al., 2020). Yet, these works tend to focus on specific settings: Pereira et al. (2018) and Okada et al. (2017) apply model predictive control to DAGGER (Ross et al., 2011) and to a Monte-Carlo-based recurrent setting respectively, while Mairal et al. (2011) and Amos and Kolter (2017) instead focus on specific problem formulations, such as solving a lasso (Tibshirani, 1996) problem or a quadratic program.

The works which are algorithmically most similar to ours are Amos et al. (2018) and Jin et al. (2020), both of which attempt to learn arbitrary dynamics in an end-to-end fashion, with some caveats. Most crucially, the techniques of both works can only learn the parameters of dynamics (and cost) functions *of which the functional form is already known*. This relies on a human expert to craft a sufficiently accurate description of the dynamics, which for complex dynamical systems can be an insurmountable task. Myriad overcomes this challenge by using neural ordinary differential equations (Neural ODEs) (Chen et al., 2018) to learn arbitrary system dynamics directly from data. Additionally, both works only consider problems that are discrete in time, and also rely on non-differentiable solvers and thus can only compute gradients at convergence via the implicit function theorem.

Furthermore, Amos et al. (2018) leverage a quadratic program solver to work with local quadratic approximations of the dynamics, as opposed to fully nonlinear dynamics. As such, their approach is unsuitable for use with learned dynamics parametrized by neural networks. In contrast, the solvers in Myriad can be applied in nonconvex settings, unlocking the use of neural network-based dynamics models.

While the approach of Jin et al. (2020) can be used with arbitrary dynamics, the authors treat system identification and optimal control in isolation of one another, while Myriad enables the user to close the loop by using gradients from control to improve system identification, and by gathering data based on the current best guess for controls. Additionally, Jin et al. (2020) rely on a neural policy network to compute controls in the optimal control setting, while Myriad instead plans controls directly from learned dynamics.

To our knowledge, while there is active and diverse research in the areas related to this work, there has remained an ongoing lack of real-world environments and plug-and-play trajectory optimization tools for a deep learning workflow. As such, we believe that Myriad, which is implemented entirely in JAX (Bradbury et al., 2018), is well-placed to bridge trajectory optimization and deep learning with its collection of real-world tasks and differentiable optimal control algorithms.

## 3 Myriad: environments and optimizers

Myriad was developed with the machine learning community in mind, and offers environments spanning medicine, ecology, epidemiology, and engineering (a full list is provided in Table 1). The repository contains implementations of various trajectory optimization techniques, including single and multiple shooting (Betts, 2010), trapezoidal and Hermite-Simpson collocation (Kelly, 2017),

and the indirect forward-backward sweep method (Lenhart and Workman, 2007). We also offer algorithms for learning dynamics, both to identify the unknown parameters of an a priori model, and to learn a black-box Neural ODE model (Chen et al., 2018).

With the exception of off-the shelf nonlinear program solvers such as `ipopt` (Wächter and Biegler, 2006) and SLSQP (Virtanen et al., 2020), every aspect of the systems and trajectory optimizers is differentiable, allowing flexible use and easy incorporation in a deep learning practitioner's workflow.

Myriad is extensible, enabling straightforward addition of new environments, trajectory optimization techniques, nonlinear program solvers, and integration methods to the repository:

- We consider *control environments* (which we also call *systems* or *environments*) specified by their dynamics function, cost function, start state, and final time. A system can optionally include a required terminal state, a terminal cost function, and bounds on the state and controls. In order to create a new system, the user can extend `FiniteHorizonControlSystem`, an abstract class defined in `systems/base.py`. We present some of these environments below; for a table of all environments and link to documentation including full environment descriptions, see Appendices A and B.1.

- A *trajectory optimizer* has an objective function, a constraint function, control and state bounds, an initial decision variable guess, and an unravel function for sorting the decision variable array into states and controls. To implement a trajectory optimizer, the user can extend the abstract class `TrajectoryOptimizer`, an abstract class defined in `optimizers/base.py`. A table of the trajectory optimizers currently available in Myriad is given in Appendix B.2.

- A *nonlinear program solver* is set to have the same function signature as those used by standard off-the-shelf solvers such as `ipopt` and SLSQP. To implement a new solver, the user can create a function with the same signature as those in `nlp_solvers/`.

Below we give an overview of some of the environments in Myriad. For a full description of the environments, see the repository documentation; a link is in Appendix A. We note that many of these environments were inspired by work of Lenhart and Workman (2007), and both the formulation and framing of such environments should be attributed to them.

**Medicine and Epidemiology**

- In **Cancer Treatment**, we want to determine the optimal administration of chemotherapeutic drugs to reduce the number of tumour cells. Control is taken to be the strength of the drug, while the cost functional is the normalized tumour density plus the drug side effects, as done by Panetta and Fister (2003). The dynamics assume that tumour cells will be killed in proportion to the tumour population size (Skipper, 1964).

- In **Epidemic**, we aim to find the optimal vaccination strategy for managing an epidemic. Control is the percentage rate of vaccination (what proportion of the population is newly vaccinated every day), while the cost functional is the number of infectious people plus a cost quadratic in vaccination effort. The dynamics follow a SEIR model (Joshi et al., 2006).

- In **Glucose**, we want to regulate blood glucose levels in someone with diabetes, following the approach of Edelstein-Keshet (1991). Control is set to be insulin injection level, and the cost is quadratic in both the difference between current and optimal glucose level, as well as in the amount of insulin used.

**Ecology and science**

- In the **Bear Populations** setting, we manage the metapopulation of bear populations in a forest and a national park within the forest, an important problem when it comes to ensuring species preservation while also avoiding bears in human-populated areas, based on the work of Salinas et al. (2005). The controls are the rates of hunting in the forest and the national park. The cost is the number of bears that exit the forest, plus a hunting cost in each location.

- In **Mould Fungicide**, we want to decrease the size of a mould population with a fungicide. Control is the amount of fungicide used, while the cost is quadratic in both population size and amount of fungicide used.

- In **Predator Prey**, we wish to decrease the size of a pest population by means of a pesticide, which acts as control. We assume that the pest population is prey to a predator in the ecosystem, which we do not wish to impact. The dynamics follow a Lotka-Volterra model, and the cost is the final prey population, plus a quadratic control cost.

### Control

We also include several classical control environments, such as **Pendulum** (the dynamics and cost of which match OpenAI Gym (Brockman et al., 2016)), **Cart-Pole Swing-Up** as presented by Kelly (2017), and **Mountain Car**, which also matches Gym (Brockman et al., 2016) except for the function describing the hill, which was changed from sinusoidal to quadratic to improve stability during Neural ODE-based system identification. While these are standard problems, we believe it worthwhile to reproduce them for study in the trajectory optimization setting. We also include other control problems such as the challenging **Rocket Landing** domain described by Açıkmeşe et al. (2013), and the forced **Van der Pol** oscillator, as presented by Andersson et al. (2019).

## 4  Trajectory optimization

Many control problems can be formulated in the language of trajectory optimization, in which an optimization technique is used to find a control trajectory which minimizes an integrated cost. While trajectory optimization approaches are rarely considered by RL practitioners, they often provide good solutions when using a known dynamics model, and thus can serve as a useful benchmark for many optimal control tasks. To give a flavour of the techniques used in trajectory optimization, here we present the standard method of direct single shooting (Betts, 2010), which is implemented in Myriad alongside other algorithms.

Letting $\boldsymbol{u}$ and $\boldsymbol{x}$ represent control and state functions, $c$ the instantaneous cost and $f$ the system dynamics, the trajectory optimization problem can be written as

$$
\begin{aligned}
\min_{\boldsymbol{u}(t)\ \forall t \in [t_s, t_f]} \quad & \int_{t_s}^{t_f} c(\boldsymbol{x}(t), \boldsymbol{u}(t), t)\, dt \\
\text{such that} \quad & \dot{\boldsymbol{x}}(t) = f(\boldsymbol{x}(t), \boldsymbol{u}(t))\ \forall t \in [t_s, t_f] \\
\text{with} \quad & \boldsymbol{x}(t_s) = \boldsymbol{x}_s \\
\text{and*} \quad & \boldsymbol{x}(t_f) = \boldsymbol{x}_f \\
\text{and*} \quad & \boldsymbol{x}_{\text{lower}}(t) \leq \boldsymbol{x}(t) \leq \boldsymbol{x}_{\text{upper}}(t)\ \forall t \in [t_s, t_f] \\
\text{and*} \quad & \boldsymbol{u}_{\text{lower}}(t) \leq \boldsymbol{u}(t) \leq \boldsymbol{u}_{\text{upper}}(t)\ \forall t \in [t_s, t_f]
\end{aligned}
\tag{1}
$$

where asterisks indicate optional constraints. Note that we allow time-dependent cost, but assume time-independent dynamics. First, we augment the system dynamics with the instantaneous cost:

$$
f_{\text{aug}}(\boldsymbol{x}(t), \boldsymbol{u}(t), t) = \begin{bmatrix} f(\boldsymbol{x}(t), \boldsymbol{u}(t)) \\ c(\boldsymbol{x}(t), \boldsymbol{u}(t), t) \end{bmatrix}.
\tag{2}
$$

Then the integral

$$
\begin{bmatrix} \boldsymbol{x}_s \\ 0 \end{bmatrix} + \int_{t_s}^{t_f} f_{\text{aug}}(\boldsymbol{x}(t), \boldsymbol{u}(t), t)\, dt = \begin{bmatrix} \boldsymbol{x}_f \\ c_f \end{bmatrix}
\tag{3}
$$

will contain the integrated cost – the objective we want to minimize – as its final entry. Let $\psi$ be a function which, given a sequence of controls and a timestamp, returns an interpolated control value.[1]

Letting $\boldsymbol{x}(t_s) = \boldsymbol{x}_s$ and $c(t_s) = 0$, we can construct the following nonlinear program (NLP):

$$
\begin{aligned}
\text{decision variables} \quad & \hat{\boldsymbol{u}}_0, \hat{\boldsymbol{u}}_1, \hat{\boldsymbol{u}}_2, \ldots, \hat{\boldsymbol{u}}_N \\
\text{objective} \quad & \left[ \int_{t_s}^{t_f} f_{\text{aug}}\left( \begin{bmatrix} \boldsymbol{x}(t) \\ c(t) \end{bmatrix}, \psi(\hat{\boldsymbol{u}}_{0:N}, t), t \right) dt \right] \texttt{[-1]} \\
\text{equality constraints*} \quad & \boldsymbol{x}_f = \boldsymbol{x}_s + \int_{t_s}^{t_f} f(\boldsymbol{x}(t), \psi(\hat{\boldsymbol{u}}_{0:N}, t))\, dt \\
\text{inequality constraints*} \quad & \boldsymbol{u}_i^{\text{lower}} \leq \hat{\boldsymbol{u}}_i \leq \boldsymbol{u}_i^{\text{upper}} \quad \text{for } i = 0, \ldots, N
\end{aligned}
\tag{4}
$$

---

[1]How this interpolation is performed depends on the integration method applied. Matching the control discretization with a fixed integration timestep circumvents the need for explicit interpolation.

To gain more intuition about direct single shooting, we visualize a toy problem of projectile motion, in which we are trying to get a projectile to an altitude of 100m after exactly 100s by choosing a launch velocity. Under simplifying assumptions, given state $\boldsymbol{x} = [x, \dot{x}]^\top$, the dynamics can be written as $f(\boldsymbol{x}) = [\dot{x}, -g]^\top$, where $g$ is gravitational acceleration. Figure 2 shows the outcome of applying direct single shooting to this problem.

Direct single shooting is the perhaps the simplest of the trajectory optimization techniques, but it comes with several shortcomings, of which we mention the two most impactful. First, direct single shooting does not allow us to impose constraints on the state trajectory found by the trajectory optimization solver. Yet, for an algorithm to be safe to apply in real-world settings, it is crucial that the user be able to restrict the system to a set of safe states (in robotics, avoiding collisions; in chemical engineering, avoiding unsafe pressure/temperature levels, etc.). Second, direct single shooting is inherently sequential, making it possibly slower and less effective in long-horizon tasks due to integration time and vanishing gradients. Other optimization techniques, such as direct multiple shooting (Betts, 2010) and direct collocation (Kelly, 2017) support parallelization, and might be much more efficient when solving over long time horizons. For a table of the trajectory optimization techniques provided in Myriad (which includes those mentioned above), see Appendix B.2.

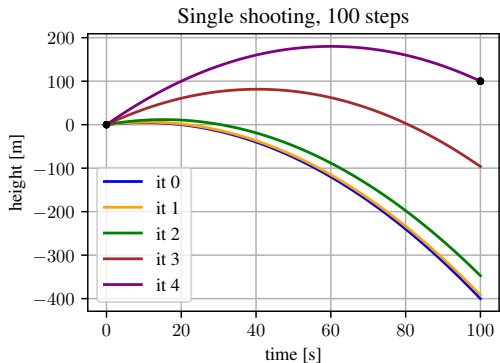

**Figure 2:** Trajectories computed after 0 to 4 iterations of direct single shooting. At each iteration, the gradient is propagated through the forward integration back to the initial parameters, which are updated to decrease the final defect.

## 5 Constrained optimization at scale

The nonlinear programs of Section 4 are usually solved using second-order techniques based on Newton's method (Boggs and Tolle, 1995; Nocedal and Wright, 2006). We would like to be able to solve these nonlinear programs at scale, leveraging the advantages that GPU-based optimization has brought to deep learning (Krizhevsky et al., 2012; Abadi et al., 2016). Unfortunately, many software implementations such as `ipopt` and `SLSQP` are restricted to CPU, instead relying on sparse matrix operations for computational efficiency (Wächter and Biegler, 2006; Virtanen et al., 2020). While effective for small models, such higher-order techniques struggle when applied over a large number of parameters, due to the size of the resulting Hessian matrix (Martens and Grosse, 2015). Indeed, since GPU computation is more suitable for dense matrix operations (Fatahalian et al., 2004), the emphasis on sparsity in traditional solvers is of little help when it comes to deep learning applications. Other problems further exacerbate the challenge of using these solvers for ML: not only do higher-order techniques tend to perform poorly in high-dimensional settings due to the prevalence of saddle-points (Dauphin et al., 2014); these solvers are also non-differentiable, making them practically impossible to use in methods where we need to propagate gradients through the solve itself (such as the imitation learning technique presented in Section 7).

Here we present a simple technique with which we have found success on many Myriad environments, and which runs fully on GPU. Let $f$ be the objective function, and $h$ the equality constraints (for example, these could be the objective and equality constraints from Eq. (4) or Eq. (14)). We use $\boldsymbol{y} = [\boldsymbol{x}, \boldsymbol{u}]^\top$ to denote decision variables of the NLP. The Lagrangian of our NLP is then

$$\mathcal{L}(\boldsymbol{y}, \boldsymbol{\lambda}) = f(\boldsymbol{y}) + \boldsymbol{\lambda}^\top h(\boldsymbol{y}), \tag{5}$$

where $\boldsymbol{\lambda}$ are known as the Lagrange multipliers of the problem. We see that a solution to the NLP will correspond to the solution of the min-max game: $\min_{\boldsymbol{y}} \max_{\boldsymbol{\lambda}} \mathcal{L}(\boldsymbol{y}, \boldsymbol{\lambda})$ (Kushner and Sanvicente, 1975). In particular, this solution will satisfy the first-order optimality condition that $(D_1 \mathcal{L})(\boldsymbol{y}^\star, \boldsymbol{\lambda}^\star) = 0$ (Bertsekas, 1999). We can attempt to find a solution by applying a first-order

Lagrangian method (Duguid, 1960; Uzawa et al., 1958) to find $(\boldsymbol{y}^\star, \boldsymbol{\lambda}^\star)$:

$$\begin{aligned}
\boldsymbol{y}^{(i+1)} &\leftarrow \boldsymbol{y}^{(i)} - \eta_{\boldsymbol{y}} \cdot (D_1 \, f)(\boldsymbol{y}^{(i)}, \boldsymbol{\lambda}^{(i)}) \\
\boldsymbol{\lambda}^{(i+1)} &\leftarrow \boldsymbol{\lambda}^{(i)} + \eta_{\boldsymbol{\lambda}} \cdot (D_2 \, f)(\boldsymbol{y}^{(i)}, \boldsymbol{\lambda}^{(i)}).
\end{aligned} \tag{6}$$

As an instance of gradient descent-ascent (Lin et al., 2020), this method can suffer from oscillatory and even divergent dynamics (Polyak, 1970). One way to mitigate this is the extragradient method (Korpelevich, 1976; Gidel et al., 2018). Instead of following the gradient at the current iterate, extragradient performs a "lookahead step", effectively evaluating the gradient that would occur at a future step. It then applies the lookahead gradient to the current iterate.

$$\begin{aligned}
\bar{\boldsymbol{y}}^{(i)} &\leftarrow \boldsymbol{y}^{(i)} - \eta_{\boldsymbol{y}} \cdot (D_1 \, f)(\boldsymbol{y}^{(i)}, \boldsymbol{\lambda}^{(i)}) \\
\bar{\boldsymbol{\lambda}}^{(i)} &\leftarrow \boldsymbol{\lambda}^{(i)} + \eta_{\boldsymbol{\lambda}} \cdot (D_2 \, f)(\boldsymbol{y}^{(i)}, \boldsymbol{\lambda}^{(i)}) \\
\boldsymbol{y}^{(i+1)} &\leftarrow \boldsymbol{y}^{(i)} - \eta_{\boldsymbol{y}} \cdot (D_1 \, f)(\bar{\boldsymbol{y}}^{(i)}, \bar{\boldsymbol{\lambda}}^{(i)}) \\
\boldsymbol{\lambda}^{(i+1)} &\leftarrow \boldsymbol{\lambda}^{(i)} + \eta_{\boldsymbol{\lambda}} \cdot (D_2 \, f)(\bar{\boldsymbol{y}}^{(i)}, \bar{\boldsymbol{\lambda}}^{(i)}).
\end{aligned} \tag{7}$$

This approach has seen recent success in the generative adversarial model literature, and it seems likely that further improvements can be made by leveraging synergies with game-theoretic optimization (Schuurmans and Zinkevich, 2016; Kodali et al., 2017; Wiatrak and Albrecht, 2019).

In practice, since we are considering real-world systems, we often want to restrict the trajectories the agent can take through state space to a safe subset. There are several ways to include inequalities when using a Lagrangian-based approach; they are described in Appendix F.

# 6 System identification

Sections 4 and 5 showed how to solve a standard trajectory optimization problem assuming known dynamics. While such techniques can be used as a basic benchmark for RL algorithms, it is often more realistic to compare an RL approach with a setting in which trajectory optimization is performed on *learned* dynamics. To this end, we turn our attention to learning system dynamics from data, i.e., the problem of *system identification* (SysID) (Keesman, 2011).

In control theory, SysID is typically performed to learn the parameters of a highly structured model developed by field experts. Indeed, such highly structured models have been used even in recent work at the intersection of learning and control (Amos et al., 2018; Jin et al., 2020). Not only is this task comparatively simple due to having to learn only a handful of parameters; in the case of identifiable systems, it is also easy to verify the accuracy of the learned model by simply checking the values of the learned parameters.

Yet the construction of a structured model relies on the ability of a human expert to accurately describe the dynamics, which is a lengthy process at best, and impossible for sufficiently complex systems. RL circumvents this issue either by not using a world model, or by building one from data (Sutton and Barto, 2018; Moerland et al., 2020). In order to provide a trajectory optimization benchmark, we must learn a model directly from data. We do this by modelling the dynamics of a system with a Neural ODE (Chen et al., 2018): a natural fit when it comes to continuous systems. While Neural ODEs have not yet been extensively studied in the context of controllable environments (Kidger et al., 2020; Alvarez et al., 2020), it is not challenging to extend them to this setting. In this case we would like to find Neural ODE parameters $\boldsymbol{\theta}$ which best approximate the true dynamics:

$$f(\boldsymbol{x}(t), \boldsymbol{u}(t), \boldsymbol{\theta}) \equiv \texttt{apply\_net}\big(\boldsymbol{\theta}, [\boldsymbol{x}(t), \boldsymbol{u}(t)]^\top\big) \approx f(\boldsymbol{x}(t), \boldsymbol{u}(t)), \tag{8}$$

where $f(\boldsymbol{x}(t), \boldsymbol{u}(t))$ is the true dynamics function. In order to train this model, consider a trajectory of states[2] $\boldsymbol{x}$, sampled with noise from the true dynamics, given controls $\boldsymbol{u}_{0:N}$. We would like our model to predict this trajectory. In particular, $\tilde{\boldsymbol{x}}$ should approximate $\boldsymbol{x}$:

$$\tilde{\boldsymbol{x}} = \left[ \boldsymbol{x}_0, \boldsymbol{x}_0 + \int_{t_0}^{t_1} f(\boldsymbol{x}(t), \psi(\boldsymbol{u}_{0:N}, t), \boldsymbol{\theta}) \, dt, \dots, \boldsymbol{x}_0 + \int_{t_0}^{t_N} f(\boldsymbol{x}(t), \psi(\boldsymbol{u}_{0:N}, t), \boldsymbol{\theta}) \, dt \right]. \tag{9}$$

---

[2] Myriad offers several methods for generating trajectory datasets, including uniformly at random, Gaussian random walk, and sampling around a candidate control trajectory.

We minimize the mean squared error between the two trajectories ($N$ is number of timesteps, $D$ is state dimension, giving $\boldsymbol{x}$ and $\tilde{\boldsymbol{x}}$ dimensions $(D, N)$). The loss is then calculated as[3]:

$$L(\hat{\boldsymbol{\theta}}) = \frac{1}{ND}\|\tilde{\boldsymbol{x}} - \boldsymbol{x}\|_E^2, \tag{10}$$

where $\| \bullet \|_E^2$ is the squared Euclidean norm (sum of squares of elements).

## 7 End-to-end SysID and control

In the Neural ODE setting, the individual parameters no longer convey an intuitive physical meaning. Yet, we can still compare the learned to the true dynamics by considering the effect of a given control sequence across a range of states. An example of such visualization is shown in Figure 3, which compares the Neural ODE learned model's dynamics with those of the true dynamics on a mould fungicide domain (Lenhart and Workman, 2007). We use automatic differentiation to calculate the gradient of the loss in Eq. 10 with respect to network parameters; another approach is to apply the adjoint sensitivity method as done by Chen et al. (2018).

As in other areas of machine learning, RL has seen increasing interest in forgoing the use of explicit models, instead structuring the policy to include a planning inductive bias such that an agent can perform *implicit* planning (Tamar et al., 2016; Deac et al., 2020; Amos et al., 2018; Jin et al., 2020). A classic example is value iteration networks (Tamar et al., 2016), which replace explicit value iteration with an inductive bias in the form of a convolutional neural network (Fukushima, 1988; LeCun et al., 1989).

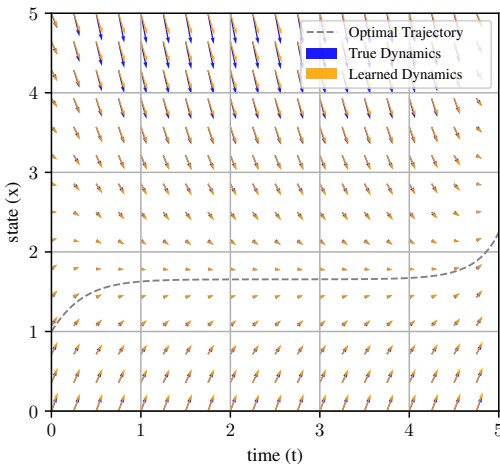

**Figure 3:** Comparison of true dynamics and learned dynamics (Neural ODE model) when applying optimal controls in the Mould Fungicide domain. We observe that the dynamics learned via SysID closely match the true dynamics for this problem.

Inspired by implicit planning, we consider a fully differentiable algorithm which performs trajectory optimization on an implicit model. By propagating gradients through the trajectory optimization procedure itself, the agent can learn directly from the loss received from acting in the real environment. In order to describe this approach – a form of "unrolled optimization" (Maclaurin et al., 2015) – we consider a modification to the Lagrangian of Eq. (5), adding parameters $\boldsymbol{\theta}$ which parametrize the underlying dynamics function. We let $\boldsymbol{y}$ represent the primal variables (control and state decision variables), $\boldsymbol{\lambda}$ the dual variables (Lagrange multipliers), $f$ the objective function, and $h$ the equality constraints. To simplify notation, we let $\boldsymbol{z} = [\boldsymbol{y}, \boldsymbol{\lambda}]^\top$, which gives the Lagrangian:

$$\mathcal{L}(\boldsymbol{\theta}, \boldsymbol{z}) = f(\boldsymbol{y}, \boldsymbol{\theta}) + \boldsymbol{\lambda}^\top h(\boldsymbol{y}, \boldsymbol{\theta}). \tag{11}$$

Let $\psi$ be a function representing the nonlinear program solver, which takes parameter values $\hat{\boldsymbol{\theta}}$ and returns $\hat{\boldsymbol{z}}$, and let $L$ be the loss function of Eq. 10. We would like to propagate the gradient of $L$ with respect to $\hat{\boldsymbol{\theta}}$ through $\psi$ at our current decision variables $\hat{\boldsymbol{y}}$. The basic procedure to achieve this is shown in Algorithm 1.

---

[3]In practice, the loss calculation is performed in parallel over a minibatch of training trajectories.

---

**Algorithm 1** End-to-End − Theory

---

1: Initialize $\hat{\boldsymbol{u}}_{0:N}, \hat{\boldsymbol{\theta}}$ with random values
2: **while** $\hat{\boldsymbol{u}}_{0:N}, \hat{\boldsymbol{\theta}}$ not converged **do**
3:     $\hat{\boldsymbol{z}} \leftarrow \psi(\hat{\boldsymbol{\theta}})$                                 ▷ solve NLP represented by Eq. (11)
4:     $\hat{\boldsymbol{x}}, \hat{\boldsymbol{u}}_{0:N}, \hat{\boldsymbol{\lambda}} \leftarrow \hat{\boldsymbol{z}}$                                     ▷ extract controls
5:     $\hat{\boldsymbol{\theta}} \leftarrow$ update using $(D\ (L \circ \psi))(\hat{\boldsymbol{\theta}})$
6: **end while**
7: **return** $\hat{\boldsymbol{u}}_{0:N}$

---

The clear challenge is the implementation of Line 5. By the chain rule we have that

$$(D\ (L \circ \psi))\ (\boldsymbol{\theta}) = (D\ L)(\psi(\boldsymbol{\theta})) \cdot (D\ \psi)(\boldsymbol{\theta}). \tag{12}$$

The first term, $(D\ L)(\psi(\boldsymbol{\theta}))$, can simply be calculated using automatic differentiation in the imitation learning setting, or using a gradient approximation method in the RL setting (Williams, 1992). The calculation of $(D\ \psi)(\boldsymbol{\theta})$ is more challenging, since it involves differentiating through the NLP solver. A natural first approach is to apply the implicit function theorem (IFT), which suggests that for $(\boldsymbol{\theta}, \boldsymbol{z})$ such that $\boldsymbol{z} = \psi(\boldsymbol{\theta})$ and $(D_1\ \mathcal{L})(\boldsymbol{\theta}, \boldsymbol{z})$ is near zero, we have

$$(D\ \psi)(\boldsymbol{\theta}) = -\ (D_2 D_1\ \mathcal{L})^{-1}\ (\boldsymbol{\theta}, \boldsymbol{z}) \cdot \left(D_1^2\ \mathcal{L}\right)(\boldsymbol{\theta}, \boldsymbol{z}). \tag{13}$$

In practice, we experienced several drawbacks when using this method. Most notably, we found the requirement that $(D_1\ \mathcal{L})(\boldsymbol{\theta}, \boldsymbol{z})$ be near zero in order for the implicit function theorem to hold particularly challenging, since an unreasonable amount of computation must be spent to achieve such high accuracy from the NLP solver.

A practical workaround is to use a *partial* solution at each timestep, and take gradients through an unrolled differentiable NLP solver. By performing several gradient updates per iteration and warm-starting each step at the previous iterate, we are able to progress towards an optimal solution with a computationally feasible approach. We reset the warm-start after a large number of iterations, as in (Jin et al., 2020), to avoid catastrophic forgetting of previously-seen dynamics. This approach, which we use in our imitation learning algorithm implementation, is presented in Algorithm 2.

---

**Algorithm 2** End-to-End Approach − Practice

---

1: Initialize $\hat{\boldsymbol{u}}_{0:N}, \hat{\boldsymbol{\theta}}$ with random values
2: **while** $\hat{\boldsymbol{u}}_{0:N}, \hat{\boldsymbol{\theta}}$ not converged **do**
3:     $\hat{\boldsymbol{z}}, \texttt{dz\_dtheta} \leftarrow$ simultaneously take several steps of $\psi(\hat{\boldsymbol{\theta}})$ and accumulate gradients
4:     $\hat{\boldsymbol{x}}, \hat{\boldsymbol{u}}_{0:N}, \hat{\boldsymbol{\lambda}} \leftarrow \hat{\boldsymbol{z}}$                                 ▷ extract controls
5:     $\texttt{dL\_dz} \leftarrow (D\ L)(\hat{\boldsymbol{z}})$         ▷ using automatic differentiation or gradient approximation
6:     $\texttt{dL\_dtheta} \leftarrow \texttt{dL\_dz} \cdot \texttt{dz\_dtheta}$                     ▷ apply the chain rule
7:     $\hat{\boldsymbol{\theta}} \leftarrow$ update with $\texttt{dL\_dtheta}$
8: **end while**
9: **return** $\hat{\boldsymbol{u}}_{0:N}$

---

We find that Algorithm 2 is able to learn effective models and propose good controls for several environments. To gain intuition about how the model learns its environment over time, we take snapshots of the controls proposed by our algorithm over the course of training. We give an example of this in Figure 4, which shows the progress of end-to-end training of a Neural ODE model on a cancer treatment domain (Lenhart and Workman, 2007).

## 8 Conclusion

Implemented in JAX, the systems and tools in Myriad fit seamlessly in a deep learning workflow, and can serve both to develop new algorithms and benchmark them against existing optimal control techniques. The current environments span medicine, ecology, epidemiology, and engineering, and

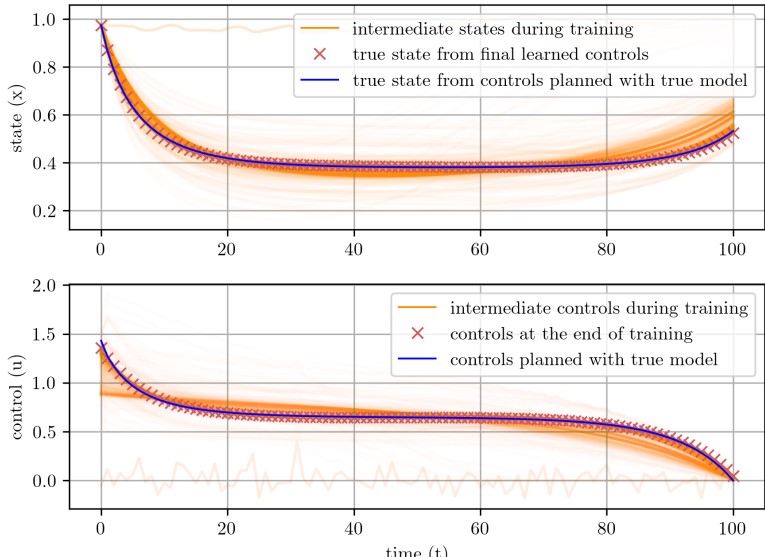

**Figure 4:** Visualization of how the controls, and corresponding states, evolve over the course of training a Neural ODE model end-to-end on the Cancer Treatment domain. The control trajectory, and corresponding state trajectory, are sampled regularly over the course of training. Each is plotted with a low alpha value to show where the learning procedure spent time during training.

special attention has been made to allow easy integration of new environments, optimizers, and nonlinear programming tools. We showcase the power of Myriad's tools by developing a novel control-oriented imitation learning algorithm which combines optimal control with deep learning in an end-to-end trainable approach. Not only does the algorithm achieve good performance on several environments; Myriad also enables comparison of this new technique with traditional trajectory optimization over fixed or learned system dynamics.

**Limitations:** There are several limitations to the Myriad repository as well as to the imitation learning algorithm developed with Myriad tools. First and foremost is the fact that many of the environments in Myriad were selected in part because they were known to be amenable to solution via traditional optimal control methods. As a result, the dimensionality of state observations in all Myriad environments is low (<10 dimensions) compared with pixel-based tasks which require representation learning of visual features (Mnih et al., 2015). Thus, Myriad is not at present useful for benchmarking the effectiveness of learned visual representations in a deep RL setting. Technical limitations are also present: for now, only fixed-step integration methods are supported, limiting our ability to take advantage of regions of simple dynamics for faster integration. Another beneficial enhancement would be the implementation of variable scaling, which would help avoid numerical stability issues which can sometimes occur when integrating through rapidly changing dynamics. Finally, an important shortcoming of the imitation learning algorithm is its inability to learn a cost function, which in the general setting is not known during learning. It would be desirable to expand the model's learning capability to include a cost function – similar to the approach of Jin et al. (2020) but using a more expressive Neural ODE model (Chen et al., 2018) – and benchmark its performance compared with the current version.

**Societal impact:** Many of the environments presented in Myriad are inspired by real-world problems (Lenhart and Workman, 2007; Betts, 2010). However, we caution that they should not by themselves be used for medical, ecological, epidemiological, or any other real-world application, since they abstract away real-world application-specific details which must be examined and approached on a case-by-case basis by experts in the domain. For example, it is important to consider the safe limits of operation and have a fallback control routine in the case of controlling robots or industrial processes. Our goal is that the Myriad testbed will help build interest within the machine learning community to bring our algorithms to application in impactful real-world settings.

**Acknowledgements:** Thank you to Lama Saouma for input regarding the feasibility of an end-to-end approach for SysID and Control and to Andrei M. Romascanu for feedback on a previous version. The authors are also thank Hydro-Québec, Samsung Electronics Co., Ldt., Facebook CIFAR AI, and IVADO for their funding, and Calcul Québec and Compute Canada for compute resources.

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
