# OpenReview forum: "Myriad: a real-world testbed to bridge trajectory optimization and deep learning"
_NeurIPS.cc/2022/Track/Datasets_and_Benchmarks — NeurIPS 2022 Datasets and Benchmarks _

### Official Review · Reviewer_naa2 · 2022-07-14
**A great library for neural continuous control**

**Rating:** 8
**Confidence:** 4
**Correctness:** The paper is technically correct.
**Clarity:** The paper is clearly written.

**Strengths:**

Most RL works in the current literature focus on complex dynamics, such as games, which has discrete timesteps, or robotics, which follows end-effect control to simplify RL learning. Althoug these RL applications are great, there remains a large space of important problems of relatively simple continuous dynamics. Trajectory optimizations have been developed for decades for these problems and it remains particularly important to answer the question whether RL algorithms can be used to further improve the state-of-the-art to these problems. There are only limited works on this domain with very limited testbeds available. I personally appreciate the efforts from the authors to make such a clear library for the RL community. It will be a great testbed for potentially huge RL advances in the continous control problems.

**Weaknesses:**

The paper is generally great. If possible, including some simple RL implementation and results would make the paper even stronger.

**Additional Feedback:**

See my comments before.

**Documentation:**

The library has good documentation.

**Relation To Prior Work:**

The discussion is sufficient.

**Summary And Contributions:**

This paper develops a library that contains a suite of real-world continuous control problems and a collection of trajectory optimization implementations, which can serve as an excellent testbed for RL practitioners for challenging control problems.

---

> ### Author Response · Authors · 2022-08-09
> **Author Response**
>
> Thank you for your review and for your idea regarding the possible inclusion of an RL implementation in Myriad and corresponding results in the benchmark scores.
>
> It is indeed our intention to have an RL implementation (most likely PPO) running in Myriad in the near future. This will serve not only as another benchmark, but also as a launching pad for testing out trajectory optimization-inspired algorithm tweaks that we hope will lead to improved performance.
>
> This said, we are unsure whether such an RL implementation will be ready in time to include results in the paper itself (and are almost completely sure that it will not be ready before the end of this review process).

---

### Official Review · Reviewer_CQCv · 2022-07-25
**Differentiable trajectory optimization library with optimal control benchmarks**

**Rating:** 7
**Confidence:** 4

**Strengths:**

The authors give a variety of benchmark problems, including those that are not usually considered by the optimal control/RL community. The included trajectory optimization algorithms should provide a good counterpart or baselines to compare against RL algorithms.

**Weaknesses:**

I’m confused what the major contribution of the paper is: is it the trajectory optimization library or is it the provided problems? If it’s trajectory optimization, there are other libraries that also provide differentiable trajectory optimization (see “Relation to Prior Work”), so this work needs to differentiate itself from the others. If it’s the included problems, they are underemphasized in the paper and need to be discussed more. Since the community is already very familiar with robotics and video game benchmarks, a discussion on these new problems (what they are, why they're interesting, aspects they have that current canonical problems don't have) can make this library a compelling choice.

**Additional Feedback:**

- The paper appears to have extra spacing, which may be due to Overleaf or compiling the document on Windows. I would suggest compiling the document on a Linux program.

- Any reason you used JAX instead of PyTorch?

While the rating is a bit low, I'm happy to increase it as long as the authors address:
- Clarifying the main message of the paper (differentiable trajectory optimization vs. the included problems) and addressing my corresponding concerns
- Helping me overcome the issues running their code out of the box.

**Clarity:**

The paper does a pretty good job of showing how the optimization algorithm works, including the pragmatic approach taken for the end-to-end training. That being said, I think Section 4 is a bit too detailed. The discussion on multiple shooting seems excessive and confused me at first when I read the paper. I think that part can be moved to the Appendix for readers that are interested in what multiple shooting is, but I think it’s okay to just go over single shooting and mention that multiple shooting additionally breaks the problem up into subproblems. This should also provide extra space to talk about the included benchmark problems.

**Correctness:**

The experiments in the paper seemed proper. I’m unable to run the code (see “Documentation”), so I cannot verify any results myself.

**Documentation:**

I’m unable to run the examples on the Github. Here’s the error I get when trying to run `python run.py --system=CARTPOLE --optimizer=COLLOCATION`:
```
Traceback (most recent call last):
  File "run.py", line 18, in <module>
    from myriad.experiments.e2e_sysid import run_endtoend
  File "/home/reviewer/tmp/myriad/myriad/__init__.py", line 5, in <module>
    from .trajectory_optimizers import *
  File "/home/reviewer/tmp/myriad/myriad/trajectory_optimizers/__init__.py", line 4, in <module>
    from myriad.trajectory_optimizers.base import TrajectoryOptimizer, IndirectMethodOptimizer
  File "/home/reviewer/tmp/myriad/myriad/trajectory_optimizers/base.py", line 14, in <module>
    from jax.ops import index_update
ImportError: cannot import name 'index_update' from 'jax.ops' (/home/reviewer/miniforge3/envs/myriad/lib/python3.8/site-packages/jax/ops/__init__.py)
```
This was ran with the version of JAX (0.3.15) that installed when I ran `pip install -r requirements.txt`. Trying an older version (0.2.13) of JAX gave different errors.


**Ethics:**

The ethics are sufficiently addressed to me.

**Relation To Prior Work:**

The comparison to prior benchmark libraries (e.g., Gym) looks good. These prior libraries focus on robotics and video games. On the other hand, Myriad includes problems from other domains like biology and ecology, which should be of interest to the community.

I’m unsure whether the authors have properly compared Myriad to related work on differentiable trajectory optimizers. In particular, [1] uses a differentiable form of iLQR, and [2] uses a differentiable form of a Pontryagin solver. While these works are mentioned and cited, it’s only in passing and without a comparison to their work. What are the shortcomings of these two works? What does your library provide compared to these two works?

[1] Amos, B., Jimenez, I., Sacks, J., Boots, B., & Kolter, J. Z. (2018). Differentiable MPC for end-to-end planning and control. Advances in neural information processing systems, 31.

[2] Jin, W., Wang, Z., Yang, Z., & Mou, S. (2020). Pontryagin differentiable programming: An end-to-end learning and control framework. Advances in Neural Information Processing Systems, 33, 7979-7992.

**Summary And Contributions:**

This paper introduces a differentiable trajectory optimization library for use in optimal control and end-to-end learning of trajectory optimization modules. The library uses JAX to solve constrained continuous-time, continuous-control optimal control problems, particularly via a scalable form of gradient descent-ascent on an associated Lagrangian. To support system identification or learnable dynamics, the library uses the neural ODE to support continuous-time evaluation while being efficient to train. The neural ODE may be trained via system identification (i.e., supervised learning on observed trajectories) or in an end-to-end fashion by backpropagating through the trajectory optimization procedure to perform imitation learning. The authors demonstrate their library using all of the aforementioned features on some proposed benchmark problems, including problems that are not usually considered by the optimal control/RL community (e.g., cancer treatment and population dynamics).

Post-rebuttal\
===========\
Raised my score from 5 to 7 after discussion.

---

> ### Author Response · Authors · 2022-08-10
> **Author Response [Part 1/2]**
>
> Thank you for your review and your helpful thoughts on how to improve the paper and repository, as well as your willingness to increase your rating.
>
> Regarding the “Weaknesses” section:
> - The main message of the paper is meant to be a mix of the two things you describe, of the kind: “look everyone, we should be working to make RL work on real-world problems, and here are some trajectory optimization methods that might be really helpful to use as a benchmark for RL (and elements of which can be incorporated in new IL/RL algorithms) since they have historically seen much success on real-world problems”.
> - Though the balance of focus is close to 50/50, if we had to choose, the provision of differentiable trajectory optimization techniques in a deep-learning compatible setting would probably count as the most important contribution. Are there any places in particular where you feel this needs to be clarified?
> - As such, we will flesh out the description in the “Related Work” section to better address the differences between this work and previous work (of which there are important differences – see our response to the “Relation to Prior Work” section for a more detailed explanation of this).
> - We will most likely include a more detailed description of the environments in Myriad in the Appendix. If not, we will at least include in the Appendix a link to the documentation, where a more detailed explanation is already present and which we will also develop further.
>
> Regarding the “Clarity” section:
> - Thank you for the suggestion to push Section 4’s explanation of multiple shooting to the Appendix and to use the leftover space to present/explain the benchmark problems. We wanted to present multiple shooting because it is less well-known than single shooting and has the important property of parallelizability, a feature which we expect to be much appreciated by ML practitioners. That said, it seems reasonable to at least move the description of the NLP arising from multiple shooting to the Appendix, and to use the resulting space for other things.
> - We will likely include (in the paper body) a presentation and explanation of (some of) the benchmark problems as per your suggestion.
> - We will definitely include a more detailed explanation of the benchmark problems (or link to it) in the Appendix (see response to the “Weaknesses” section above).
>
> [continued in next comment]

---

> > ### Author Response · Authors · 2022-08-10
> > **Author Response [Part 2/2]**
> >
> > [continuing from previous comment]
> >
> > Regarding the “Relation to Prior Work” section:
> >
> > - Thank you for raising this point. In fact, this work was originally inspired by “Differentiable MPC” (Amos et al., 2018). “Pontryagin Differentiable Programming” (PDP) (Jin et al., 2020) has also served as a useful reference during the project.
> > - While we mention these works in Section 6 (SysID), we agree that it would be beneficial to also address them explicitly in Section 2 (Related Work).
> > - There are several important distinctions between our work and those mentioned:
> >     - In (Amos et al., 2018):
> >     - The authors only learn a handful of parameters used to describe the environment dynamics. While Myriad also supports this “SysID” functionality, we also support a much more general form of dynamics learning via Neural ODEs. In particular, they explore Cartpole and Pendulum, for which Myriad is able to learn the parameters using SysID on <10 example trajectories (of course, Myriad takes much more than this to learn dynamics when using a Neural ODE-based approach).
> >     - The authors consider a quadratic approximation to the dynamics, whereas our solver works on the fully nonlinear dynamics (true or learned).
> >     - The authors mention that their approach often fails when using neural networks to represent dynamics, because it’s a non-convex problem. Myriad does not suffer from this theoretical limitation (in practice, we do find that it is hard to get solver convergence when using dynamics parametrized by a neural network. That said, our end-to-end approach does not require convergence in order to learn a useful model).
> > - In (Jin et al., 2020):
> >     - The main contribution is, to our understanding, a (re)framing of the three related but disparate problems of SysID, Optimal Control, and Inverse Optimal Control into one general mathematical framework. Myriad is less theoretical/more practical in nature, instead focusing on giving researchers trajectory optimization tools which they can use in a deep learning setting, as well as environments to test algorithms on.
> >     - The authors treat SysID and Optimal Control as tasks which occur separately, whereas Myriad allows multiple ways of “closing the loop” between SysID and Optimal Control.
> >     - The authors use a non-differentiable approach (ipopt) to compute the Optimal Control solution, whereas we also provide approaches which are fully differentiable.
> >     - In the Optimal Control setting, the authors use a neural policy to calculate controls, while Myriad instead uses the learned dynamics to plan optimal controls.
> > - In both papers:
> >     - The authors consider control problems that are discrete in time, while Myriad is built to handle continuous-time dynamics out-of-the-box, and can be used with different integration methods and different timestep sizes.
> >
> > Regarding the “Documentation” section:
> >
> > - Thank you for bringing this error to our attention; indeed, it seems a recent update to JAX broke things. The reason forcing an older version of JAX also didn’t work is to our understanding because of an interaction with Python’s (modern) chex and the old version of JAX.
> > - We have updated the code to conform to the style of the current version of JAX (0.3.15), so if you try following the instructions again it should work out-of-the-box.
> > - We also uncovered an issue in the interaction between the (updated) parsing libraries, and have pushed a somewhat quick fix to this for now, but intend to address this issue more cleanly moving forward.
> >
> > Regarding the “Additional Feedback” section:
> >
> > - The paper was compiled in Overleaf. We would be open to compiling locally (on MacOS or possibly Linux) and seeing if it changes anything. Could you please tell us where you see the extra spacing?
> > - JAX vs PyTorch:
> >     - There are several reasons behind this choice:
> >     - When this project started (2019), PyTorch did not yet have (and indeed to some extent still lags JAX) sufficiently versatile automatic differentiation capabilities.
> >     - We wanted to write environments that were fully differentiable. JAX enables this in a very straightforward way.
> >     - We also like the straightforward parallelization that JAX offers.
> >     - We expect that JAX will see increasingly widespread adoption in the machine learning community in the years to come.
> >
> > Please let us know if there is anything we can provide more clarification on or help with.

---

> > > ### Comment · Reviewer_CQCv · 2022-08-26
> > > **Response**
> > >
> > > Hi authors,
> > >
> > > Thank you for your detailed feedback and effort in responding to my concerns. I can confirm that the examples on the Github now successfully run on my machine. I believe enough of my concerns have been addressed to bump the score.
> > >
> > > A couple other notes:
> > > - Line 60: "describes two standard trajectory optimization techniques" should be revised since multiple shooting is moved to the Appendix.
> > > - Your revision has put the references and checklist in the supplementary section when they should be in the main paper.
> > > - Regarding the spacing issue, if you compile the document with the `[preprint]` option (so that it is `\usepackage[preprint]{neurips_data_2022}`), you should see the amount of spacing change. I've found that this spacing issue for the submission option happens on Overleaf.
> > >
> > > After looking into the documentation of the included systems, I noticed the dynamical system equations seem simple and many systems are described by a first-order ODE over a scalar $x(t)$. On the other hand, most canonical benchmark problems that use differential equations under the hood (e.g., MuJoCo, Brax) use high-dimensional systems like robots with many links. This makes the systems much more challenging and interesting to control.
> > >
> > > Since Myriad's included systems appear "simple," I'm not sure if Myriad has been demonstrated to work on problems that would seem more realistic.
> > > - Am I mistaken about the included systems? That the system equations seem "compact" makes them seem "simple." Is there something I'm missing that actually makes the included systems "difficult?"
> > > - Can Myriad be extended to work with [Brax](https://github.com/google/brax) (the simulator based on JAX)? I'm aware that Brax is discrete-time, but I believe they perform Euler integration on the underlying ODEs. If that's the case, it should be possible to utilize the integration techniques your paper covers. This avenue should make Myriad very appealing to the robotics community.

---

> > > > ### Author Response · Authors · 2022-08-28
> > > > **Thank you for your additional feedback**
> > > >
> > > > Thank you for your additional response, for your helpful pointers, and for your bumped score.
> > > >
> > > > - We have uploaded another revision which corrects the typo in line 60 and also puts references and checklist in the correct place. We will continue to update the paper tomorrow (improving spacing and perhaps slightly lengthening some explanations with the extra space).
> > > > - We have switched to using TexLive 2020, which appears to mitigate the spacing issue you brought up (see the very last point in this page: https://neurips.cc/Conferences/2022/PaperInformation/NeurIPS-FAQ).
> > > > Thank you for bringing both of these to our attention.
> > > >
> > > > Regarding your comment on the complexity of the included systems:
> > > >
> > > > - It is true that seven of the systems contain a one-dimensional state variable. These are perhaps over-represented in the examples given in the paper, since they are easy to visualize.
> > > > - The remaining eleven systems have higher-dimensional state variables (though admittedly lower-dimensional than some of the tasks in MuJoCo and Brax).
> > > > - Many of the systems are “difficult” (or at least, not “easy”) for other reasons:
> > > >     - Two of the systems (bear populations, rocket landing) have a multi-dimensional control.
> > > >     - 13 of the systems have nonlinear dynamics (of which seven have dynamics which are neither linear nor quadratic).
> > > >     - Six systems have a fixed terminal state (as in, the system must be in that configuration at the end of the episode).
> > > >     - Three systems (bacteria, predator prey, tumour) have a terminal cost which is applied only at the end of the trajectory, based on the terminal state.
> > > >     - Two systems (harvest, timber) have time-dependent cost.
> > > >     - Three systems have a cost which is linear in control, which can make the optimization process more challenging since the optimal solution often involves discontinuities (Lenhart and Workman, 2007, Chapter 17).
> > > >
> > > > Regarding the question whether the techniques in Myriad work on “realistic” tasks, we would note that real-world (and simulated) versions of Pendulum and Cart Pole are readily solved by trajectory optimization techniques. Indeed, the trajectory optimization techniques presented in Myriad are widely used in industry, aerospace, and engineering (see Lenhart and Workman’s “Optimal Control Applied to Biological Models”, Betts’ “Practical Methods for Optimal Control and Estimation Using Nonlinear Programming”, and Biegler’s “Nonlinear Programming: Concepts, Algorithms, and Applications to Chemical Processes”). All this said, we agree it would be interesting to include systems with very complex dynamics in Myriad, and explore the performance of trajectory optimization on such problems.
> > > >
> > > > Regarding the Brax question:
> > > >
> > > > - We agree that it would be very interesting to run the techniques in Myriad on the systems presented in Brax.
> > > > - We also understand that Brax uses Euler integration.
> > > > - After a brief initial investigation, it unfortunately does not seem obvious how to use the Brax environments in Myriad. This is due to the fact that, by default, Myriad uses dynamics equations, written in the form of ODEs, to describe a system. In Brax, to our understanding, a system is defined by specifying a “system config” (see, for example, from Line 256 in https://github.com/google/brax/blob/c06fd32c34d34a238a57923d849b64de148066ee/brax/envs/hopper.py) and letting the physics engine work out the stepping details. It appears to us that the integration step is built into the physics engine, and it does not appear obvious how to extract the derivative from this.
> > > > - If you have any insight into how this could be made to work, we would like to hear it!

---

> > > > > ### Comment · Reviewer_CQCv · 2022-08-29
> > > > > **Response**
> > > > >
> > > > > Hi authors,
> > > > >
> > > > > Thanks for your detailed response. It seems to me the repo is full of many appealing features:
> > > > > - different forward integration techniques,
> > > > > - variety of trajectory optimization algorithms (including many differentiable ones), and
> > > > > - a fair basis of included problems, including in domains not normally considered by the RL/control community
> > > > > It also tackles the continuous-time domain whereas many prior works only consider discrete time.
> > > > >
> > > > > It remains to be seen how useful the included problems are since they are predominantly low-dimensional. Furthermore, unlike physics simulators, each system must have the dynamics equations explicitly provided by the user, rather than merely a system configuration file (e.g., MJCF).
> > > > >
> > > > > Altogether, however, the provided tools and paper seem well-put-together enough for me to recommend an accept.
> > > > >
> > > > > I would highly recommend the authors keep pushing on the functionality and capabilities of Myriad so that the community is more like to adopt it. Some suggestions:
> > > > > - Reach out to the Brax developers to see if it is possible to hook Brax into Myriad.
> > > > > - Study what systems that Myriad can work on. Does it work on systems with discontinuous dynamics (e.g., locomotion systems)? High-dimensional systems? Can it handle learning latent dynamics from images [1]?
> > > > >
> > > > > As a side note, I cannot run the notebooks since they cannot find `myriad`. This is despite me being able to do that in the Python interpreter.
> > > > >
> > > > > [1] Hafner, Danijar, Timothy Lillicrap, Jimmy Ba, and Mohammad Norouzi. "Dream to control: Learning behaviors by latent imagination." arXiv preprint arXiv:1912.01603 (2019).

---

> > > > > > ### Author Response · Authors · 2022-08-29
> > > > > > **Thank you for your additional response**
> > > > > >
> > > > > > Thank you for your additional response and for the again-bumped score.
> > > > > >
> > > > > > We have updated the README with specific instructions for how to run the jupyter notebooks. In conjunction with this, we have updated the repo so that Myriad can be built and installed (for example, in the same venv as the requirements.txt). Following the new instructions should fix the error you were getting (which we also reproduced when running from outside the IDE).
> > > > > >
> > > > > > Thank you for your suggestions on how to keep pushing the functionality of Myriad. Compatibility with Brax in particular is a possibility we are excited to explore.
> > > > > >
> > > > > > Finally, thank you for your insights in general during the review period. Your feedback was very helpful and is much appreciated.

---

### Official Review · Reviewer_hCWc · 2022-07-27
**Review of Myriad: a real-world testbed to bridge trajectory optimization and deep learning**

**Rating:** 5
**Confidence:** 3
**Correctness:** Please see details in the weakness se…
**Clarity:** Please see details in the weakness se…

**Strengths:**

- Benchmarking of trajectory optimization algorithms, and making them differentiable is relevant yet under-explored to the ML community, and this paper provides a good starting point towards filling the gap.
- The experiments span an interesting range of benchmarking trajectory optimization algorithms with true dynamics, dynamics learning, and trajectory optimization with learned dynamics.
- The repository seems well structured and documented


**Weaknesses:**

- Details of the tasks are missing. There is only a single-sentence description of the tasks in the appendix, which makes it difficult to tell if these tasks are really useful and proper for benchmarking trajectory optimization algorithms. For example, the paper claims the tasks to be challenging, but I am not sure if tasks such as pendulum, mountain car, and Cart-Pole Swing-Up can be considered challenging, which are more like standard easy-to-solve benchmarking tasks. I am less familiar with other tasks in medicine, ecology, and epidemiology, and thus not sure if those tasks are really challenging and can be used for benchmarking trajectory optmization methods. Please provide more details of the tasks and the reasons for choosing them.
- The paper is not very well structured and miss some important details:
Section 4, I don't think such a detailed explanation and comparison between the direct single shooting and direct multiple shooting is necessary (including figure 2), since they are just existing trajectory optimization methods and should not be the focus of the work. I feel this section, could be changed to a discussion on how different trajectory optimization methods work on the **proposed tasks**, in stead of how they work in a toy example as in current figure 1.
Section 5, it seems that the proposed method to do constraint optimization is just to do gradient-descent-ascent on a GPU. I am not sure why this would just be significantly faster than traditional methods well optimized on a CPU. is there any specific optimization used for running it on the GPU? Or at least some quantitative comparison of the speed can be provided to understand if the proposed method is much faster and can be viewed as a contribution.
Besides, how is the trajectory optimizer such as direct single shooting made differentiable? Is it, e.g., just by unrolling the computation graph under a automatic differentiable framework (JAX), or does it leverage some other techniques like implicit function theorem? This is unclear from the current paper.
- Some statements are not accurate: the abstract states that the paper "enables machine learning researchers to benchmark imitation learning and reinforcement learning algorithms against trajectory optimization-based methods in challenging real-world environments". But later in the limitation section the paper states "it is at present unlikely to be useful for benchmarking RL tasks ...". Please be consistent on the statements. Also, from Table 4 in the appendix, most of the tasks can be defined with just a few (<10) parameters, which seems to indicate these tasks are not really "challenging".


**Additional Feedback:**

Please see details in the weakness section.

**Documentation:**

Yes

**Relation To Prior Work:**

Yes.

**Summary And Contributions:**

This paper presents a suite of tasks inspired from real-world problems. It also provides implementations of 5 trajectory optimaiztion algorithms. The environments and the trajectory optimization algorithms are made differentiable. This paper also demonstrates system identification and dynamcis learning via neural ODE on the presented tasks. It also proposes a new imitation learning algorithm that embeds planning into the policy structure by leveraging the differentiating through the non-linear program solver. Finally, references scores of the trajectory optimization with true dynamics, sysID-ed dynamics, learned dynamics, and the proposed imitation learning algorithms are reported.

---

> ### Author Response · Authors · 2022-08-10
> **Author Response**
>
> Thank you for your review and for your thoughts on how to improve the paper.
>
> - “Details of the tasks are missing. …”
>     - Currently a more detailed explanation of the tasks is present in the repository’s documentation, but we will further improve it and also include some or all of it in the Appendix. We will likely also include a simplified explanation (at least of several of the environments) in the main paper body.
>     - It is true that most of the “continuous control” tasks (pendulum, mountain car, cart-pole swing-up) are somewhat easy to solve in trajectory optimization and RL. As such, we view them as a good starting point for testing whether new algorithms perform as hoped/expected. We would also like to note that many of the other Myriad environments have not typically been seen in either trajectory optimization or RL settings, and as such can be interesting to study. While some of these environments (Mould Fungicide, Cancer Treatment) are relatively straight-forward to solve in a trajectory optimization setting, others (Timber Harvest, Rocket Landing) have optimal solutions which are difficult to find. Furthermore, in initial experiments we find that even some of the systems with simpler dynamics (like Cancer Treatment) are nontrivial to solve with basic RL methods. This brings up the interesting question of whether trajectory optimization can be more effective than RL in some settings, and as RL practitioners, how we should modify our algorithms and perspective to take advantage of this.
>
> - “The paper is not very well structured and…”
>     - Thank you for the suggestion regarding multiple shooting. We wanted to present multiple shooting because it is less well-known than single shooting and has the important property of parallelizability, a feature which we expect to be much appreciated by ML practitioners. That said, it seems reasonable to at least move the description of the NLP arising from multiple shooting to the Appendix, and to use the resulting space for other things. In particular, we intend to use some of this space for an explanation of (at least some of) the benchmark tasks.
>     - We use the Extragradient method to solve the nonlinear program resulting from the trajectory optimization problem. Extragradient is a Lagrangian-based method and indeed similar to GDA. Both Extragradient and GDA are implemented in Myriad and run on GPU. It is in fact not faster than traditional higher-order CPU-optimized methods; indeed, in many cases it takes both more iterations and more wall-clock time to find a solution. The important aspect of these algorithms is that they are implemented entirely in JAX, and thus can run on GPU and be differentiated via loop unrolling or via forward accumulation of gradients (the latter being what we do in practice). Do you think the paper would benefit from an explanation like the one given above?
>     - The idea of using the implicit function theorem is explored in limited detail in Section 7 when we talk about the end-to-end implicit planning imitation learning algorithm presented in Algorithm 1. In fact, the initial goal was to use high-performance non-differentiable trajectory optimization solvers (like ipopt and SLSQP) off-the-shelf, and use the implicit function theorem to propagate gradients. Unfortunately, this came with many challenges, the most problematic being that one needs to wait until the solver is sufficiently close to a fixed point in order to apply the IFT, which for many tasks takes prohibitively long to be useful.
>     - We note that implementing more nuanced differentiable NLP solvers (such as that presented in http://users.iems.northwestern.edu/~nocedal/PDFfiles/semore.pdf) could have a very positive impact on the speed of our methods.
>
> - “Some statements are not accurate…”
>     - Thank you for bringing up these points.
>     - We do believe some of the tasks to be difficult (especially for RL approaches). The “at present unlikely” sentence (which we will change to make more clear) was meant to highlight the fact that we don’t support pixel-based control, so Myriad isn't at present useful to benchmark algorithms which have a vision-based representation learning component.
>     - We would note that parameter count is not necessarily a good measure of “task complexity” (of Kolmogorov complexity it could be a good measure, but we care more about the difficulty of solving the task). We do agree that the SysID problem is a relatively straightforward one given the low parameter counts and mostly identifiable systems. But unlike past papers in this area (Differetiable MPC by Amos et al., 2018; Pontryagin Differentiable Programming by Jin et al., 2020), we are interested in learning full Neural ODE models of the dynamics, which is a much harder and more general problem than learning just the handful of parameters that describe the true system dynamics. Is there somewhere in particular that would benefit from this note in your opinion?

---

### Official Review · Reviewer_tBG5 · 2022-07-28
**A partial bridge between trajectory optimization and deep learning.**

**Rating:** 5
**Confidence:** 4
**Clarity:** The presentation and writing are of v…

**Strengths:**

Deep learning-based approaches to decision making are not often compared to relevant benchmark results from more classical, continuous-time trajectory optimization, even when such comparisons are possible in principle. Myriad helps bridge this gap by means of a new repository and results presented in the paper. The challenges and the solutions taken by Myriad are explained with care, particularly in the key case where system dynamics need to be learned. The main example of such a learned model is a differentiable Neural ODE, outlined in Algo 2, for which experimental results are provided.

The possibility of combining deep learning and trajectory optimization methods is intriguing.

Several of Myriad’s current limitations (like its inability to handle pixel-based images) are clearly highlighted.

**Weaknesses:**

It’s not clear to a deep RL practitioner like me how to use Myriad’s results as benchmarks against which to meaningfully compare RL or IL techniques.

Environments called cart-pole and pendulum (wrapped in the standard gym API) do exist. But I see no reason to conclude that their underlying equations of motion are at all the same as the systems of the same name covered by Myriad. If they are the same, then the paper should document this correspondence, and compare Myriad’s results in those environments to appropriate RL or IL results, even if only from prior work.

But if the underlying dynamics of Myriad’s systems are not the same as those of existing gym environments, then Myriad should bridge the gap (between continuous time and discrete time) by implementing gym environments which do share those dynamics. This would provide the missing part of the bridge between deep learning and optimal control that Myriad is targeting.

**Additional Feedback:**

A description of the amount of compute required to produce the paper’s experimental results would be very useful.

**Correctness:**

I’m not able to assess the correctness of the paper’s formal descriptions of Myriad’s approach to continuous-time trajectory optimization, but I see no problem with the results presented.

**Documentation:**

The repository appears to be comprehensive and well-documented.

**Ethics:**

No concerns.

**Relation To Prior Work:**

Short but solid discussion of prior work.

**Summary And Contributions:**

Myriad is a testbed consisting of 18 continuous-time, optimal control problems (called systems or environments) from several real-world domains (medicine, ecology, and epidemiology). This work aims to enable comparisons between deep learning methods (like RL and IL) and optimal control methods, for which benchmark reference scores are provided. Written in JAX, Myriad also enables new algorithms like implicit planning over neural ODEs, a novel control-oriented imitation learning technique that is presented in detail.

---

> ### Author Response · Authors · 2022-08-10
> **Author Response**
>
> Thank you for your review and for your ideas on how to make Myriad more immediately accessible to RL practitioners.
>
> Regarding the “Weaknesses” section:
>
> - “It’s not clear to a deep RL practitioner (...) how to use Myriad’s results”:
>     - This point is well-taken. We wanted to take an approach with Myriad which is different from the heavily structured nature of (for example) the OpenAI Gym environments. In particular, Myriad is intended to give maximum freedom to the user, allowing for them to choose details such as integration method, environment time horizon, number of controls to solve for and interpolate between, etc. Of course, this freedom comes with the flipside that the specification and implementation of such details falls on the user: in particular, there is not currently a Gym-like API into which an RL algorithm can be directly plugged (for better or worse, this is similar to how practically all real-world applications by default do not have a “nice” API available, and it is similarly up to the practitioner to make design choices about how exactly to model the problem). The advantage of this freedom is that the practitioner has the possibility to use techniques which are impossible with a standard API-based approach.
>     - Having said this, we do plan to implement an RL algorithm (likely PPO) for these tasks in the near future, and to do so will require creation of an API at some level of abstraction, which will of course be made available to plug into other RL algorithms (though as per the above point, we would not like this to become the only way that RL practitioners interact with the testbed). That said, we are unsure whether this RL implementation will be ready in time for this paper, and are almost certain that it won’t be ready by the end of the review window.
> - “Environments called cart-pole and pendulum…”
>     - Indeed environments of these names do exist in many places, and you have good reason to not implicitly assume their equations of motion match those of OpenAI Gym. In the case of pendulum, the equations of motion exactly match those in Gym’s classic_control suite (https://github.com/openai/gym/blob/master/gym/envs/classic_control/pendulum.py), though they are at present somewhat arbitrarily slowed by a factor of 20 so that we can use a “reasonable” time horizon of 15. In the case of cart-pole swing-up, the equations of motion are slightly different from those provided in Gym (note also that the Myriad cart-pole environment is the swing-up problem, not the balancing problem provided in Gym), following instead those presented in https://epubs.siam.org/doi/epdf/10.1137/16M1062569 and elsewhere. Note that the mountain car environment is also custom, though heavily inspired by the OpenAI Gym one (in Myriad the “valley” function was changed to avoid numerical divergence issues encountered when learning a Neural ODE dynamics model of the original sinusoidal one).
> The idea of comparison with prior IL/RL results is an interesting one which we could in theory do for the pendulum task. Do you have any recommendations of a standard fixed-horizon benchmark of the pendulum task that we could reference?
> - “But if the underlying dynamics…”
>    - We very much agree that it would make the RL practitioner’s life simpler to include a Gym API for the Myriad environments (and as mentioned in a previous comment, having an RL implementation running across Myriad environments is a main task on the roadmap ahead). While the availability of a Myriad RL API should exist in the near future, as we mentioned before, we would like to emphasize that the ethos of Myriad is also about keeping the complexity and richness of real-world problems (such as continuous-time dynamics, terminal (and even non-terminal) state constraints, terminal costs, etc.), and bringing these challenges and considerations into the realm of RL research.
>
> Regarding the “Additional Feedback” section:
> - We will include a description of the (approximate) amount of compute used for each of the four benchmarks in the Appendix. As a ballpark figure, all of the trajectory optimization problems can be solved in seconds to minutes, SysID can be done in <30mins for each environment, Neural ODE dynamics learning (whether end-to-end or not) can be done in <3h for each environment (possibly shorter depending on the environment). These numbers come from running on a personal laptop with a quad core i7, 16GB of RAM, and integrated graphics.

---

> > ### Comment · Reviewer_tBG5 · 2022-08-20
> > **Thank you for the clear responses**
> >
> > The author response states that unlike “heavily structured” OpenAI Gym environments, “Myriad is intended to give maximum freedom to the user, allowing for them to choose details such as integration method, environment time horizon, number of controls to solve for and interpolate between”. While options are useful, they could be included as configurable settings in a Gym-like API. At the very least, the paper should give a concrete example of an actual comparison of Myriad with RL and/or IL. Without this, Myriad falls short of its goal of enabling “machine learning researchers to benchmark imitation learning and reinforcement learning algorithms against trajectory optimization-based methods".
> >
> > I look forward to the Myriad RL API which is currently planned. Until that API is available, I believe the proposed “bridge” is too partial to be of much use to the NeurIPS community.

---

### Official Review · Reviewer_vypE · 2022-07-28
**Good work to fill the blank of hands-on tool for the area**

**Rating:** 9
**Confidence:** 2
**Correctness:** Yes.
**Clarity:** Yes.

**Strengths:**

This paper offered trajectory optimization tools to the machine learning community that is compatible to deep learning workflow. It encorages the development of machine learning algorithms with the goal of addressing real-world problems.

**Weaknesses:**

The tasks that Myriad provides is simple and thoroughly studied in optimal control, while ML for trajectory optimization usually aims to solve more complex tasks.

**Additional Feedback:**

No.

**Documentation:**

Yes.

**Ethics:**

No.

**Relation To Prior Work:**

Yes.

**Summary And Contributions:**

This paper proposed Myriad, which is a real-world testbed that offers many real-world relevant, continuous space and time dynamical system environments for optimal control. Myriad is written in JAX, and both environments and trajectory optimization routines are fully differentiable.

---

> ### Author Response · Authors · 2022-08-09
> **Author Response**
>
> Thank you for your review and for your feedback on the repository environments.
>
> It is true that most of the “continuous control” tasks (pendulum, mountain car, cart-pole swing-up) are somewhat easy to solve in trajectory optimization and reinforcement learning. As such, we view them as a good starting point for testing whether new algorithms perform as hoped/expected.
>
> We would also like to note that many of the other Myriad environments have not typically been seen in either trajectory optimization or RL settings, and as such can be interesting to study. While some of these environments (Mould Fungicide, Cancer Treatment) are relatively straight-forward to solve in a trajectory optimization setting, others (Timber Harvest, Rocket Landing) have optimal solutions which are difficult to find. Furthermore, in initial experiments we find that even some of the systems with simpler dynamics (like Cancer Treatment) are nontrivial to solve with basic RL methods. This brings up the interesting question of whether trajectory optimization can be more effective than RL in some settings, and as RL practitioners, how we should modify our algorithms and perspective to take advantage of this.

---

### Official Review · Reviewer_qJTb · 2022-07-31
**Myriad: a real-world testbed to bridge trajectory optimization and deep learning**

**Rating:** 6
**Confidence:** 3
**Correctness:** I agree the claims made in the submis…

**Strengths:**

1. All tasks are inspired by real-world problems, with applications in medicine, ecology, and epidemiology.
2. Myriad is, as the authors mentioned, the first repository that enables deep learning methods to be combined seamlessly with traditional trajectory optimization techniques.
3. The system dynamics in Myriad are continuous in time and space, offering several advantages over discretized environments.
4. The authors present a novel control-oriented imitation learning algorithm that combines optimal control with deep learning.

**Weaknesses:**

1. As a real-world testbed paper, I think it will be more helpful to have tools like Jupiter notebooks to directly demonstrate some examples mentioned in the paper. I went through the GitHub page mentioned in the paper. The code is clean and nice but it will be better to have some demonstrations.
2. The paper is well organized but it seems the authors are trying to put too much content in relatively short pages, which make it harder to captcha more details in each section, in this case, the different scenarios.

**Additional Feedback:**

Overall I recommend accepting this paper if we have enough room. The Github page can be better with more demo and details for each section is necessary.

**Clarity:**

The paper is well written. But it will be better to have more details in each section, rather than mentioning section 4567 together in the paper. I also recommend that, since you have a github page, you should make it for better use. For example, you can put each section in the github and with a small demo for each scenario.

**Documentation:**

Yes, this paper has a github page, but according to my previous comment, it could be better to have more details and some demo jupyter notebook pages as well.

**Ethics:**

I find no concern for this paper.

**Relation To Prior Work:**

Yes

**Summary And Contributions:**

The authors present Myriad, a testbed written in JAX which enables machine learning researchers to benchmark imitation learning and reinforcement learning algorithms against trajectory optimization-based methods in challenging real-world environments. Myriad contains 18 optimal control problems presented in continuous time and ranging from biology to medicine to engineering. As such, Myriad strives to serve as a stepping stone toward the application of modern machine learning techniques for impactful real-world tasks.

All environments, optimizers and tools are available in the software package at https://github.com/nikihowe/myriad.

---

> ### Author Response · Authors · 2022-08-09
> **Author Response**
>
> Thank you for your review and for your ideas on how to improve the paper and repository. Regarding the points you raise:
>
> - As per your suggestion, we will improve the explanation of the examples given in the repository README.
> - Regarding your GitHub suggestions, we will write short examples/explanations (possibly in the README, possibly only in Jupyter notebooks which we will point to from the README – see next comment) for how to run each of the Sections 4, 5, 6, 7 of the paper.
> - We will implement example Jupyter notebooks as you suggest. We think that for most cases, this will involve little more than pointing to scripts that are currently in the "experiments" directory. We will also improve the documentation of the scripts in that directory. We intend for the scripts (both in "experiments" and in run.py) to be easy to interact with, especially when used in conjunction with the examples we’ll have for the different Sections. Does this sound in line with your hopes?
> - Regarding your comments about there being a lot of material in 4, 5, 6, 7: it is true that we had a lot of content to fit into 9 pages. We wanted to make sure we covered the capabilities of the repository, and the unique things one can do with it compared with standard testbeds. Please let us know if there is a specific section which you think could be clarified, and/or if there is a section which you found less useful than the others. At this point, the most likely change is that we cut some details from Section 4 and give it to other sections.

---

### Author Response · Authors · 2022-08-26
**First revision in response to review feedback**

We have uploaded a revised version of the paper which incorporates most of the feedback we received in the reviews (thank you all).

Specifically, this version, which is not intended to be final, includes the following changes:
- Significantly expanded the "related work" section, paying particular attention to the comparison of this work with Differentiable MPC and Pontryagin Differentiable Programming, two important previous works which attempt to solve problems similar to those presented in Myriad.
- Provided a presentation of several (11) of the environments in Myriad in Section 3.
- Included a section in the Appendix describing the amount of compute used for each benchmark task, as well as the total amount of compute used for all experiments.
- Moved the discussion of Direct Multiple Shooting to the Appendix.
- Included a sentence in the Appendix pointing the reader to the repository documentation for a detailed description of all the environments.

Additionally, we have created Jupyter notebooks for the tasks described in sections 4, 5, 6, 7 -- see the “examples” directory in the project repository to run these guided examples.

In the coming days, we will also:
- Decide whether or not to bring detailed descriptions of the environments to the Appendix. At present these only exist in the documentation of the repository. We are still deciding whether to simply include a pointer to this documentation in the Appendix, or to include the descriptions directly, which would be more convenient for the interested reader but would also significantly lengthen the Appendix.
- Improve the repository README to better introduce and guide the user through the Myriad codebase.

---

### Meta-Review · Area_Chair_9AmT · 2022-09-09

**Recommendation:** Accept
**Confidence:** 3

**Metareview:**

This paper provides a test-bed called Myriad for trajectory optimization and system ID in jax, with the hope of engaging RL practitioners to benchmark against the test bed. The proposed test-bed provides examples ranging from different domains (such as medicine, and biology) deviating from traditional domains that are usually the focus on RL benchmarks. Further, the focus is on continuous time settings. Limitations are clearly specified. Authors have done a good job of describing the testbed and comparing existing methods.

One challenge that remains to be addressed is how useful this testbed really is for RL practitioners. This concern has been raised because it seems like the dynamics of the proposed problems fall on the simpler end of the spectrum. Nonetheless, I am currently of the belief that having a JAX based test-bed for such domains is still valuable and I am hoping will contribute more to reproducible RL results in these domains. Currently it appears that the designer makes a lot of choices around the design and set up of the RL framework. It may be a real concern if the general RL community will not adopt the testbed for this precise reason. More importantly, it did strike me as odd that the contribution and abstract claim testbed for imitation learning and RL but do not provide a simple example of how this could be done. This was the main concern of tBG5. I also agree with CQCv's assessment that requiring dynamics equations to be explicitly provided by the user will significantly cost adoption of the test-bed. Reviewer vypE scored the paper very well but failed to justify the score for me to rely significantly on it.

My expectation is that the authors will genuinely deliver on all the asks. Further, continue to improve the library to make it more amenable to testing of RL algorithms with a more friendly API. Overall I want to note that significant effort seems to be required to put together the test-bed but also believe all above concerns are valid and authors are highly recommended to incorporate as many changes as possible. The lack of explicit examples of simple baseline rl testing on the test-bed is concerning, irrespective of the API and potential simplicity of the domains (which I believe is not a huge concern if it warrants RL testing in novel domains). I strongly encourage the authors to incorporate feedback and I believe it will make for a much stronger testbed in practice. Hoping authors deliver on this, to the extent possible by camera-ready deadline and after, I am recommending an accept since the testbed has some utility even in its current form (though I am less optimistic about widespread adoption in its current form).

---

### Decision · Program_Chairs · 2022-09-16

Accept